# Pro-Inflammatory Serum Amyloid a Stimulates Renal Dysfunction and Enhances Atherosclerosis in Apo E-Deficient Mice

**DOI:** 10.3390/ijms222212582

**Published:** 2021-11-22

**Authors:** Antony Gao, Sameesh Gupta, Han Shi, Yuyang Liu, Angie L. Schroder, Paul K. Witting, Gulfam Ahmad

**Affiliations:** Redox Biology Group, Discipline of Pathology, Faculty of Medicine and Health, Charles Perkins Centre, The University of Sydney, Sydney, NSW 2006, Australia; agao2399@uni.sydney.edu.au (A.G.); sameesh.gupta@student.unsw.edu.au (S.G.); hshi2945@uni.sydney.edu.au (H.S.); anna.liu@sydney.edu.au (Y.L.); asch7719@uni.sydney.edu.au (A.L.S.); paul.witting@sydney.edu.au (P.K.W.)

**Keywords:** serum amyloid A, atherosclerosis, renal, dysfunction, pro-inflammatory

## Abstract

Acute serum amyloid A (SAA) is an apolipoprotein that mediates pro-inflammatory and pro-atherogenic pathways. SAA-mediated signalling is diverse and includes canonical and acute immunoregulatory pathways in a range of cell types and organs. This study aimed to further elucidate the roles for SAA in the pathogenesis of vascular and renal dysfunction. Two groups of male ApoE-deficient mice were administered SAA (100 µL, 120 µg/mL) or vehicle control (100 µL PBS) and monitored for 4 or 16 weeks after SAA treatment; tissue was harvested for biochemical and histological analyses at each time point. Under these conditions, SAA administration induced crosstalk between NF-κB and Nrf2 transcriptional factors, leading to downstream induction of pro-inflammatory mediators and antioxidant response elements 4 weeks after SAA administration, respectively. SAA treatment stimulated an upregulation of renal IFN-γ with a concomitant increase in renal levels of p38 MAPK and matrix metalloproteinase (MMP) activities, which is linked to tissue fibrosis. In the kidney of SAA-treated mice, the immunolocalisation of inducible nitric oxide synthase (iNOS) was markedly increased, and this was localised to the parietal epithelial cells lining Bowman’s space within glomeruli, which led to progressive renal fibrosis. Assessment of aortic root lesion at the study endpoint revealed accelerated atherosclerosis formation; animals treated with SAA also showed evidence of a thinned fibrous cap as judged by diffuse collagen staining. Together, this suggests that SAA elicits early renal dysfunction through promoting the IFN-γ-iNOS-p38 MAPK axis that manifests as the fibrosis of renal tissue and enhanced cardiovascular disease.

## 1. Introduction

The vascular endothelial monolayer serves as an interface between components of the circulating blood and the tissues in which it supplies; as such, the endothelium plays important roles in normal vascular homeostasis [1]. In response to various physical and chemical stimuli, the endothelium can modulate and regulate vascular tone, thrombotic activity, extravasation of leukocytes during infection or inflammation, and oxidative stress through the release of appropriate mediators [2]. Thus, endothelial dysfunction represents a systemic pathology linked to cardiovascular and renal diseases [1,3]. The early stages of atherogenesis are associated with an inflammatory response driving the remodelling and permeabilisation of the vascular endothelium. Monocyte adherence to the activated endothelium and transmigration into subendothelial spaces lead to the formation of lipid-laden foamy macrophages [4], which exacerbate the inflammatory response and promote atheroma formation by recruiting and further stimulating macrophage formation [5]. Calcification of atherosclerotic lesions can also occur, particularly amongst advanced lesions [6]. The expansion of atherosclerotic lesions can lead to lumen stenosis and acute plaque rupture that promotes the occlusion of smaller branching blood vessels [7]. In addition, endothelial dysfunction, as a result of inappropriate inflammatory activity, negatively impacts the highly vascularised renal system whose functionality is dependent on endothelial integrity [8]. Renal dysfunction is believed to precede various chronic systemic inflammatory diseases such as obesity, metabolic disease, atherosclerosis and cardiovascular disease due to the highly conserved nature of the inflammatory response [9,10,11].

The acute phase apolipoprotein serum amyloid A (SAA) has been increasingly utilised as a supplementary biomarker for in vivo inflammation [12]. SAA is predominantly of hepatic origin, synthesised in response to the presence of pro-inflammatory cytokines and becomes associated with high-density lipoproteins (HDLs) via exchange with the HDL Apo-AI protein upon entry into circulation. Only a small unbound portion of SAA is active; however, this portion of free SAA increases in the presence of pro-inflammatory cytokines [13,14]. Extrahepatic sources of SAA synthesis can include atherosclerotic plaques, smooth muscle cells and macrophages [15,16]. SAA can elicit pro-inflammatory activity via two main pathways: a receptor-mediated pathway, binding formyl peptide receptor-like 1 (FPRL-1), receptor for advanced glycation end products (RAGE) and toll-like receptors (TLR); or via an oxidative stress pathway involving increased production of reactive oxygen species (ROS) [17,18]. These pathways converge on and lead to the production of a suite of pro-inflammatory cytokines, which are then able to sustain the SAA response through a positive feedback mechanism. If the SAA response is inappropriately extended, the response becomes pathological, exacerbating chronic inflammation, which can then lead to pathologies such as kidney injury [19], rheumatoid arthritis [20] and diabetes mellitus [21].

SAA-induced pro-inflammatory and pro-thrombotic activities are regulated by the canonical nuclear factor kappa-light-chain-enhancer of activated B-cell (NF-κB) transcription factor, onto which both the receptor-mediated and ROS pathways converge [17,18]. Attempts at SAA receptor blockade have exhibited only variable effectiveness compared to its intrinsic association with HDL [17], and interindividual variability of HDL bioactivity presents further problems against mitigating SAA-induced pathophysiology [22]. Blockade of NF-κB activation was shown to be effective, though given the ubiquitous nature of NF-κB and its centralisation within redox, inflammatory and immunological cascades, the utilisation of such inhibitors could elicit dangerous side-effects when they act as immunosuppressants [10,23,24,25,26]. Therefore, greater understanding is required concerning the cytokines and mediators induced by SAA that contribute to downstream acute inflammation and the pathogenesis of endothelial dysfunction and long-term systemic inflammation.

Linkages between the signalling pathways that underpin SAA-induced inflammation and the induction of endothelial dysfunction remain unclear. SAA induces various inflammatory cellular responses through NF-κB activation [17], and one of the main downstream effects of NF-κB signalling is T-cell activation and the secretion of immunostimulatory cytokine interferon gamma (IFN-γ), which regulates both the activation of macrophages and major histocompatibility complex (MHC) expression, thus implicating both divisions of the immune system [27,28]. The activation of macrophages by IFN-γ elicits pro-inflammatory responses through secretion of a suite of cytokines such as tumour necrosis factor alpha (TNF-α) and pro-inflammatory interleukins (IL), which feed into a positive feedback mechanism to potentiate inflammation. Mature macrophages can also engulf and destroy pathogenic material through ROS production, which enhances oxidative stress stimuli [29].

IFN-γ, along with other pro-inflammatory cytokines (e.g., IL-1 and TNF-α) can activate inducible nitric oxide synthase (iNOS) that catalyses the conversion of L-arginine to nitric oxide (NO) [30]. While constitutive levels of iNOS remain contested, elevated iNOS is believed to be associated with high oxidative stress, with unregulated NO production triggering apoptosis and both vascular and tissue remodelling events or is converted to the potent oxidant peroxynitrite (ONOO^−^) via superoxide reaction [31,32,33]. The reactivity of ONOO^−^ with certain biomolecules can affect the integrity of endothelial cells, smooth muscle cells, as well as myocardial tissue, which may manifest in the long term as chronic inflammatory disease and organ fibrosis.

NF-κB activation can also upregulate the expression of monocyte chemoattractant protein-1 (MCP-1) in mesangial cells within renal tissue. MCP-1 is an important chemokine that mediates the infiltration of monocytes from glomerular capillaries into Bowman’s space. MCP-1, binding to the CCR2 chemokine receptor, has been shown to increase arterial lipid deposition and enhance monocyte adhesion, which can lead to inflammatory and fibrotic renal disease and accelerate atherosclerotic progression [34]. As a result of these varied responses to oxidative stress, a better understanding of the transcription factors and the downstream mediators implicated in SAA-induced oxidative stress can support further study of efficacious therapeutic targets for a wide range of inflammatory conditions. In the present study, inflammation was induced by SAA administration to investigate the role of SAA in promoting renal and vascular dysfunction in a murine model. Herein, we further elucidate the key players involved in renal inflammation as well as their long-term endpoints.

## 2. Methods

### 2.1. Murine Model

All studies involving mice were conducted with appropriate ethics approval (AEC approval 2018/1408). Male C57/BL (7 weeks old) apolipoprotein E-deficient (ApoE^−/−^) mice (Animal Resource Centre, Perth, WA, Australia) were acclimated for one week at the Charles Perkins Centre Animal Facility prior to the commencement of study. Mice were randomly housed in groups of 6 animals per cage on a 12 h light–dark cycle at 22 °C with a standard chow diet (cat#23200-12152, Specialty Feeds, Glen Forrest, WA, Australia) and water provided ad libitum.

### 2.2. Experimental Groups

At 8 weeks of age, ApoE^−/−^ mice were randomly allocated into two treatment groups: vehicle control (n = 6) and SAA groups (n = 6). Studies for each group were conducted in parallel as follows.

(a) Vehicle control group: Mice received 100 µL phosphate-buffered saline (PBS as vehicle control) via intraperitoneal (*i.p*.) route every 3 days for 14 days.

(b) SAA group: Mice received 100 µL of filter-sterilised SAA (120 µg/mL stock) administered via *i.p.* route every 3 days for a period of 14 days (equivalent to 12 µg SAA per animal per 3-day dose). This dose of administered SAA equates to maintaining a moderate level of this circulating acute phase protein that can rise up to 1000 µg/mL in plasma during a severe acute phase response [35]. The experimental design together with the planned analyses is schematically summarised in Figure 1. Separate groups of control and SAA-treated mice were monitored for 4 or 16 weeks after cessation of SAA stimulation and organs were harvested for biochemical and histological/immunohistochemical analyses at these times. Notably, these time points were selected to mimic the earliest stages prior to marked development of atherosclerotic lesions, and then at the endpoint, where more severe atherosclerosis is noted in ApoE-deficient mice.

### 2.3. Live Animal Imaging

Enhanced action of matrix metalloproteinases can initiate processes linked to tissue/organ remodelling. To assess whether SAA-mediated renal inflammation elicits increased MMP activity, mice were administered a commercial MMPsense probe 4 weeks after stimulation with SAA (2 nmol, 150 µL) via tail vein prior to imaging. At 24 h post treatment with MMPsense, animals were imaged using an IVIS^®^ SpectrumCT (PerkinElmer, Waltham, MA, USA) as described in our previous report [36]. Accumulated in vivo MMPsense signals were quantified using standard Living Image^®^ (PerkinElmer) standard data analysis software.

Following imaging, mice were sacrificed using isoflurane and death confirmed via cervical dislocation. Biological specimens, including both kidneys and urine samples, were harvested and processed for biochemical and histological analyses. Heart, blood and the entire aortic tree were also collected to study changes in vascular disease as previously described [37,38]. In a parallel chronic study, mice were randomly allocated into the same aforementioned treatment groups, vehicle control (n = 8) and SAA (n = 6), and housed in groups of 4 mice per allocated group. At 16 weeks following SAA treatment (equivalent of 18 weeks husbandry), these mice (now all 26 weeks of age) were sacrificed as described above, and their plasma, urine and organs harvested.

### 2.4. Urine Collection

Urine samples were collected from conscious mice prior to deep anaesthesia by being neck-scuffed with a capped tube (2 mL, Eppendorf, Sigma-Aldrich, Sydney, Australia) simultaneously placed over the penis in one motion. This method of collection generally yielded 100–200 µL of urine for biochemical analysis. In some mice, urine was directly collected from the bladder during the harvesting process using a 1 mL syringe with a fine gauge needle. Urine collections were immediately snap frozen in liquid nitrogen and stored at −80 °C for subsequent biochemical analysis.

### 2.5. Collection of Kidney Specimens

Upon excision, the left kidneys were assigned to histological analyses and were fixed ex vivo in 4% *v*/*v* formalin and subsequently embedded in paraffin. Kidney samples were stored as paraffin blocks at 22 °C until required. For biochemical analyses, kidney samples were snap frozen in liquid nitrogen and stored at −80 °C for subsequent tissue homogenisation (below).

Paraffin blocks containing kidney tissue were sectioned (5 µm) using a rotary microtome onto pre-marked Superfrost Plus™ slides (Thermo Fisher Scientific, Waltham, MA, USA) before being dewaxed and rehydrated. Sections were subsequently immunostained for immunohistochemistry (IHC) or immunofluorescence (IF) studies. For imaging analyses, a heat-induced antigen retrieval method was employed to enable the binding of antibodies to target proteins. Tissue sections embedded with paraffin were baked in an oven at 60 °C for 1 h to ensure adherence to the slides and then dewaxed with xylene and rehydrated with a graded alcohol series (100, 95, 70% *v*/*v*) prior to water. rehydrated slides were placed in pH 9 Target Retrieval buffer (Dako, Sydney, Australia) in the dark. Heat-retrieval was performed using a Biocare Decloaking Chamber (Dako Cytomation, Pacheco, CA, USA) and allowed to run at optimised settings. Zen 2 Lite software (Carl Zeiss, Sydney, Australia) was utilised for microscope image capture where relevant. Image analyses and other image measurements were performed on ImageJ software (v.2.0, NIH, Bethesda, MD, USA).

### 2.6. Kidney Homogenisation

Frozen right kidney tissues were thawed then cut transversely into two equal portions. One half was then minced with scissors, snap frozen once again in liquid nitrogen, then pulverised with a mortar and pestle. The remaining tissue was returned to storage at −80 °C. Roughly pulverised renal tissue was warmed to 22 °C then suspended in lysis buffer (50 mM PBS, pH 7.4 comprising 1 mM EDTA, 10 µm butylated hydroxytoluene and a Protease Inhibitor Cocktail tablet (Roche Diagnostics, Bern, Switzerland)). Next, the suspended tissues were transferred into a Teflon-reinforced glass tube and homogenised using a matching rotating Teflon-coated piston (Wheaton Specialty Glass, Millville, NJ, USA; 500 r.p.m.) on ice. After 5 min, the renal homogenates were centrifuged (2500× *g*) and the clarified supernatant was aliquoted into tubes (100 µL) for storage at −80 °C for subsequent biochemical analysis.

### 2.7. Biochemical Assays

For all biochemical analyses, total protein concentration was assessed using bicinchoninic acid (BCA) assays with serially diluted bovine serum albumin as protein standard (Sigma-Aldrich, Sydney, Australia). To account for variation in tissue wet weight, total homogenate protein was used to normalise all determined quantitative biochemical parameters.

### 2.8. Assessment of Renal Injury Biomarkers

Under normal conditions, expression levels of kidney injury molecule-1 (KIM-1) are relatively low, though inflammatory states and other renal insults can cause marked increases in KIM-1 expression within renal proximal tubule epithelial cells; as such, KIM-1 levels have been utilised as a specific biomarker for kidney injury [39]. Levels of renal KIM-1 were determined using a single-plex sandwich enzyme-linked immunosorbent assay (ELISA) kit according to instructions of the manufacturer (Abcam, Melbourne, Australia). Absorbance readings were taken at 450 nm with an Infinite^®^ M1000 Microplate reader (Tecan, Männedorf, Germany) and analysed in Microsoft Excel using an appropriate standard curve. All values were normalised using total protein levels obtained from the BCA assay.

Total urinary protein was also assessed as a marker for proteinuria and impaired glomerular function. Total urinary protein was measured using a BCA assay (above). Absorbance readings were taken at 562 nm with an Infinite^®^ M1000 Microplate reader (Tecan, Männedorf, Germany) and values were analysed in Microsoft Excel and calculated using an appropriate standard curve.

### 2.9. Assessment of Interferon-Gamma (IFN-γ) Content

The concentration of IFN-γ cytokine within kidney homogenates was determined using a commercially available ELISA kit (elisakit.com (accessed on 29 October 2021) Australia), following the instructions of the manufacturer. Briefly, kidney homogenate samples were centrifuged to separate insoluble matter (12,000× *g*, 10 min, 4 °C) before loading onto a 96-well plate along with relevant blank solutions and standards. Next, biotin-labelled detection primary were added to appropriate wells and incubated (22 °C, 2 h). Subsequently, streptavidin-HRP conjugates were added and further incubated (45 min, 37 °C). Wells were adequately washed and aspirated as recommended. Finally, TMB substrate was aliquoted to each well and the reaction mixture incubated at 22 °C in the dark. After 15 min, absorbance readings were taken at 450 nm with an Infinite^®^ M1000 Microplate reader (Tecan, Männedorf, Germany) and analysed in Microsoft Excel. The final IFN-γ concentration was calculated via interpolation from an appropriate standard curve and normalised to total homogenate protein.

### 2.10. SDS-PAGE and Western Blot Studies for p-p38 MAPK

Where required, levels of protein expression in kidney homogenates were validated and semi-quantified with Western blot analysis. Raw homogenates were centrifuged (12,000× *g*, 10 min, 4 °C), and the resultant clarified supernatants (20 µg total protein/sample) were loaded onto a 12% *w*/*v* acrylamide gel (BioRad, Sydney, Australia) and separated by electrophoresis (150 V, 30 min). Following protein separation, gels were activated with UV light for 5 min using a Gel Doc XR+ molecular imager (BioRad, Sydney, Australia). Images obtained from UV activation were later used to assess total protein load. Proteins were transferred from the gel onto a 0.2 µm Trans-Blot^®^ Turbo™ Midi polyvinylidene difluoride (PVDF) membrane transfer pack (BioRad, Sydney, Australia) utilising a semi-dry transfer method in the Trans-Blot^®^ Turbo™ Transfer System (BioRad, Sydney, Australia) for 7 min. Membranes were blocked in 5% *v*/*v* skim milk solution at 22 °C for 45 min with gentle shaking then washed with 1× TBS-T (1× TBS, 0.01% *v*/*v* Tween-20, 2 × 5 min). Subsequently, membranes were incubated overnight, with gentle agitation at 4 °C with anti-p-p38 MAPK primary antibody diluted (1:1250 *v*/*v*) in 5% BSA *w*/*v*/1× TBS-T before washing with 1× TBS-T (3 × 5 min). Membranes were then incubated with a diluted (1:10,000 *v*/*v*) polyclonal anti-rabbit IgG peroxidase-conjugated secondary antibody (Sigma-Aldrich, Sydney, Australia) for 1 h at 22 °C with gentle agitation. Finally, membranes were washed with 1× TBS-T (4 × 5 min) before addition of ECL Clarity (BioRad, Sydney, Australia) and imaging. Where required, membranes were imaged using the ChemiDoc Touch imaging system (BioRad, Sydney, Australia) as per the manufacturer’s guidelines. Densitometric data were analysed using ImageLab™ software (v 6.0.1, 2017, BioRad, Sydney, Australia). Protein expression levels were normalised against total protein load and expressed as relatives compared to controls.

## 3. Immunofluorescence (IF) Studies

Deparaffinised and rehydrated 5 µm thick renal sections first underwent heat-induced antigen retrieval (above). Slides were then incubated with 3% *v*/*v* H_2_O_2_/water for 10 min to block endogenous peroxidase activity then with serum-free protein block (Dako, Sydney, Australia) for 30 min to reduce non-specific immunostaining. Sections were subsequently incubated with a primary antibody (selected from the following: anti-phospho-NF-κB p65, Sigma-Aldrich, dilution 1:100 *v*/*v*; anti-Nrf2, Abcam, dilution 1:700 *v*/*v*; anti-IFN-γ, Abcam, dilution 1:3000 *v*/*v*; anti-phospho-p38 MAPK, Cell Signalling Technologies, dilution 1:800 *v*/*v*) for 1 h before incubating with an HRP-labelled polymer conjugated secondary antibody (Dako, Sydney, Australia) for 30 min. Slides were then incubated with an Opal fluorophore for 5 min in the dark (dilution 1:50 *v*/*v*; PerkinElmer, Melbourne, Australia). To visualise individual nuclei, slides were incubated with DAPI (dilution 1:80 *v*/*v* 0; PerkinElmer, Melbourne, Australia) for 10 min. Similarly to IHC, slides were incubated (22 °C, humidity chamber) and subject to appropriate washing steps after each reaction. Slides were mounted with fluorescence mounting medium and cover slipped for storage at 4 °C in moist and dark conditions where required. A ZEISS Axio Scope.A1 upright fluorescent microscope fitted with a digital camera (Zen 2 Lite software, Carl Zeiss, Sydney, Australia) was utilised to capture JPEG formatted images. Random fields of view were obtained from both cortical and medullary regions and fluorescence intensity was quantified using ImageJ (public freeware, NIH, USA).

### 3.1. Immunohistochemistry (IHC) Studies

To assess and semi-quantify the spatial distribution of renal iNOS enzyme, sections of kidney tissue were employed for immunohistochemical analysis. Heat-induced antigen retrieval steps (above) were completed on 5 µm deparaffinised and rehydrated thin renal sections. To block endogenous peroxidase activity, slides were incubated (10 min 3% *v*/*v* H_2_O_2_/water) and then with a ready-to-use serum-free protein block (Dako, Sydney, Australia) for 30 min to reduce non-specific immunostaining. Slides were then incubated with polyclonal anti-iNOS primary antibody (final dilution 1:200 *v*/*v*; Abcam, Melbourne, Australia) for 1 h. Subsequently, slides were incubated with a ready-to-use horseradish peroxidase (HRP)-labelled polymer conjugated secondary antibody (Dako, Sydney, Australia) for 30 min. Slides were incubated in a humidity chamber at 22 °C and all sections were washed between each reaction step. Slides were visualised using 3,3′-diaminobenzidine (DAB; Dako, Sydney, Australia) with a Harris haematoxylin counterstain. Slides were mounted and cover slipped before imaging on a ZEISS Axio Scope.A1 light microscope fitted with a digital camera (Zen 2 Lite software Carl Zeiss, Sydney, Australia). For each section, random fields of view were imaged from both cortical and medullary regions and converted to JPEG format. For iNOS, DAB-staining was quantified using ImageJ (freeware NIH, USA).

### 3.2. Assessment of Tissue Fibrosis with Picrosirius Red Staining

Renal and aortic sections were prepared from tissues isolated at 16 weeks post SAA treatment (termed the chronic murine model) and were used to assess fibrotic renal changes following SAA administration. Briefly, deparaffinised and rehydrated 5 µm renal sections were first stained with Harris haematoxylin for 10 min, washed with running water, then stained with picrosirius red solution for 1 h before washing with acidified distilled water. Subsequently, slides were dehydrated with absolute alcohol (3 × 2 min), cleared with xylene before being mounted and cover slipped. Slides were imaged using a ZEISS Axio Scope.A1 light microscope fitted with a digital camera (Zen 2 Lite software, Carl Zeiss, Sydney, Australia). Random fields of view were imaged from both cortical and medullary regions and converted to the JPEG format. Finally, picrosirius staining intensity quantified using ImageJ (freeware, NIH, USA).

## 4. Histological Assessment of Atherosclerotic Lesion Size and Composition

Hearts and the aortic arch were also collected from mice allocated to the chronic model (that corresponds to 16 weeks post cessation of SAA treatment) to study long-term aortic lesion formation following SAA administration. Briefly, heart tissues were dissected halfway through the heart and perpendicular to the aorta. Dissected heart tissues were fixed with ethanol (70% *v*/*v* ethanol/water) then dehydrated with a graded series of ethanol and embedded in paraffin. Where required, the aortic sinus tissues were sectioned, deparaffinised and rehydrated before undergoing picrosirius red staining as described above. Only aortic sections which displayed a three-valve leaflet morphology (judged by microscopic inspection) were used for further quantitative and qualitative analysis. Slides were imaged using a ZEISS Axio Scope.A1 light microscope fitted with a digital camera (Carl Zeiss, Sydney, Australia). Using Zen 2 Lite software, images were converted to JPEG format for quantitative analysis in ImageJ (freeware NIH, USA). Percentage lesion area was calculated at 5× magnification by manually outlining lesion sites for total lesion area which was then normalised to the total area of the aortic root. Additional lesion histological analyses at the aortic were conducted at 20× magnification.

## 5. Statistical Analysis

All statistical analysis was performed using GraphPad^®^ Prism Version 8.0 (GraphPad Software Inc., La Jolla, CA, USA). Group differences were compared using a parametric independent samples t-test. Graphical data were expressed as relative means ± standard deviation (SD). Differences between relative means were taken to be statistically significant at a 95% confidence interval (*p* < 0.05) and have been indicated where appropriate.

## 6. Results

### 6.1. SAA Administration Stimulate MMPsense in Mouse Kidneys

MMPs are zinc-dependant enzymes possessing proteolytic activity against extracellular matrix proteins. MMPs have been found associated with renal dysfunction particularly by causing renal fibrosis [40]. In our studies, MMPsense administration to mice 4 weeks after the cessation of SAA treatment showed increased bioluminescent signal intensity in the lower thoracic region and between the hind limbs of the mice (Figure 2A), indicating enhanced MMP activity in response in SAA treatment. This signal was largely absent in control mice (without SAA) although MMPsense activation was noted in the tail region for mice administered MMPsense, which is consistent with the recruitment of inflammatory cells that release MMPs at the site of injection injury. However, it was difficult to confirm if the in vivo bioluminescent signal detected in SAA-treated mice resulted specifically from increased renal MMP activity. To clarify, mice were sacrificed, and the isolated kidneys subsequently reimaged with IVIS^®^ SpectrumCT (PerkinElmer). Consistent with the increase in the MMP bioluminescence in SAA-treated mice being linked to renal remodelling (Figure 2B), isolated kidneys from SAA-stimulated mice showed markedly enhanced bioluminescence, particularly in the medullary region (2–3-fold higher intensity than control). Quantification of the in vivo bioluminescent signal (in the thoracic and lower gut regions) using Living Image^®^ (PerkinElmer) showed a trend towards increasing in mice administered SAA compared to the respective control, whereas the corresponding quantification of MMP activity in isolated kidneys showed a significant difference.

### 6.2. SAA Administration Causes Increased Expression of KIM-1 and Increased Secretion in Total Urinary Proteins

Under normal conditions, kidney injury molecule-1 (KIM-1) mRNA and protein are expressed at relatively low levels but can be upregulated, particularly in the proximal tubules during acute renal injury [41]. Expression of the KIM-1 biomarker is correlated with both renal and heart disease [41,42]. Kidney tissue homogenates from mice administered SAA showed a ~2-fold increase in KIM-1 expression compared to the vehicle control, although this did not reach statistical significance (Figure 3A). Consistent with the increased level of KIM-1 signifying renal injury, total urinary protein also increased ~1.3-fold higher in mice assigned to the SAA group than those in the corresponding vehicle control group (Figure 3B). While group differences for KIM-1 expression and total urinary protein did not reach statistical significance, SAA administration appeared to stimulate renal injury and a loss in selective glomerular filtration. Taken together with the outcome from MMPsense imaging, the collective data suggest that renal tissue is a target for SAA-mediated damage/remodelling, and subsequent analyses to determine the molecular basis for this injury is warranted.

### 6.3. SAA Administration Stimulates Tissue Antioxidant Response

Nuclear factor erythroid 2-related factor 2 (Nrf2) is a nuclear transcription factor that regulates the expression of antioxidant proteins in response to oxidative damage elicited by injury, inflammation or other internal metabolic events [43]. To assess the expression of Nrf2 and antioxidant activity in response to SAA-induced oxidative stress, immunofluorescence studies were conducted to localise and quantify Nrf2 expression in the mouse kidney (Figure 4A). Compared to control, renal cortical Nrf2^+^ immunostaining (red) was significantly increased in mice administered with SAA. Nrf2 staining also appeared to be largely localised to nuclei in the renal proximal tubule (Figure 3A; white arrows) with some cytoplasmic staining evident. Interestingly, glomerular Nrf2+ immunostaining was relatively low compared to tubular staining (Figure 4A; green arrows), suggesting that endothelial cells were not involved in SAA-mediated Nrf2 regulation for oxidative stress, and that expression and transcriptional activation of Nrf2 was focal to epithelial cells.

To examine inter-group differences for Nrf2 expression with greater detail, immunofluorescence images were quantified using Fiji software (ImageJ, version 2.0). Cortical Nrf2^+^ immunostaining (Figure 4B) was ~8.8 fold greater in kidney sections from the SAA group than corresponding tissue from control (*p* < 0.01). This would suggest that SAA-induced oxidative stress upregulates Nrf2 expression to further stimulate antioxidant response elements to counterbalance reactive oxygen species within the kidney, possibly by infiltrating or resident macrophages [44]. Likewise, Nrf2 immunostaining intensity was significantly higher (>20-fold) in the renal medulla in SAA-treated group compared to the control (Appendix A).

### 6.4. SAA Administration Stimulates Phospho-NF-kB p-65 Expression in the Renal Tissue

SAA-induced NF-kB signalling in pro-inflammatory activity has been previously shown, with the SAA receptor-mediated pathways and SAA-induced ROS production both converging on the phosphorylative activation of the canonical transcription factor [10,45]. Overall cortical p-p65 NF-kB expression in kidneys from mice in the SAA treatment group was greater than the corresponding vehicle control group and largely localised to the nuclei and cytoplasm of tubular epithelial cells with only some cytoplasmic NF-kB staining evident in these same cells (Figure 5A; white arrows). Similar to Nrf2, NF-kB-p65^+^ immunostaining was relatively low within the glomerular endothelial cells across all groups (Figure 4A; green arrows), suggesting that NF-kB activation predominantly occurred in cells of the tubular network. When quantified, NF-kB-p65^+^ immunostaining in the cortical region increased significantly ~3.4 times greater in the SAA group than the corresponding control (*p* < 0.01). This suggests that SAA elicited both expression and activation of Nrf2 and NF-kB were upregulated in parallel. Likewise, NF-kB immunostaining intensity was significantly higher (5-fold) in the renal medulla in SAA-treated group compared to the control (Appendix A).

### 6.5. SAA Administration Stimulates p-38 MAPK in Renal Tissue

Activation of the p38 MAPK pathway leads to a variety of biological effects, such as apoptosis and cellular differentiation and fibrosis of the kidney, liver and lungs [46]. Through a macrophage-mediated response, p38 MAPK activation stimulates immune cell activation, leading to the clearance of pathogenic and foreign material [47,48]. The functional linkage between p38 MAPK and NF-kB through COX2 signalling activation has been implicated in the reduction of Leydig cell testosterone production and in human airway myocytes as a result of chronic inflammation and oxidative stress [49,50]. Some studies have purported p38 MAPK as an upstream regulator of NF-kB, though both p38 MAPK-independent and p38 MAPK-dependent NF-kB activation pathways exist in pro-inflammatory responses. Therefore, immunofluorescence studies on p38 MAPK activation were also conducted.

Cortical p-p38 MAPK expression was significantly higher in renal sections from the SAA group compared with vehicle controls, with immunostaining largely localised to the nuclei of the tubular cells with some cytoplasmic staining (Figure 6A; refer to white arrows). As with the other immunofluorescence studies, glomerular p-p38 MAPK immunostaining was relatively low (Figure 6A; green arrows), suggesting that SAA predominantly induced p-p38 MAPK activity within tubular epithelial cells rather than within endothelial cells. Quantification of p-p38 MAPK^+^ immunostaining (Figure 6B) showed that the fluorescence signal in the renal cortical region was significantly elevated by ~ 2.5-fold in the SAA group compared to the vehicle control (*p* < 0.01). Consistent with the data determined in the cortex, p-p38 MAPK immunostaining intensity was higher (~1-fold) in the renal medulla in SAA-treated group compared to the control (Appendix A). These results suggest that SAA-stimulation activated p38 MAPK-dependent NF-kB pro-inflammatory signalling cascades.

To validate the p-p38 MAPK immunostaining data, Western blot analyses were conducted and yielded an immune-positive band identified at 38 kD (Figure 6C), consistent with anticipated molecular weight for p-p38 MAPK. Densitometric measurements of the protein bands at 38 kD followed by normalisation to the corresponding total protein load in each lane indicated that the intensity of the bands from the SAA group were ~2.5-fold greater than the vehicle controls; however, these semi-quantitative measurements did not reach statistical significance.

### 6.6. SAA Administration Stimulates IFN-γ Expression in the Renal Tissue

IFN-γ is a cytokine predominantly synthesised by natural killer cells and T cells [51,52]. IFN-γ limits tissue destruction for acute insults and has important functional and regulatory linkages with NF-kB and p38 MAPK [53,54]. Herein, we show that cortical IFN-γ content was significantly higher in the SAA group compared to vehicle controls. IFN-γ^+^ immunostaining was localised mainly to tubular epithelial cell nuclei (Figure 7A; refer to white arrows) with low-level staining detected within the glomerular parietal epithelia (Figure 7A; green arrows). Quantification of cortical IFN-γ^+^ immunostaining (Figure 7B) showed a significant ~2.7-fold increase in renal sections from the SAA group compared to controls (*p* < 0.01). Independent quantification of the IFN-γ^+^ using a commercial ELISA kit indicated that the concentration of this chemokine was ~1.8-fold higher in the SAA group than in the corresponding vehicle control (Figure 7C), although this difference was not statistically significant. Similarly, IFN-γ^+^ immunostaining intensity was significantly elevated (2.5-fold) in the renal medulla in SAA-treated mice (Figure 4).

### 6.7. SAA Administration Stimulates Inducible iNOS Expression in the Renal Tissue

iNOS is part of a family of nitric oxide synthases which catalyse the formation of nitric oxide from L-arginine; iNOS expression is linked to the chemokine IFN-γ. Renal iNOS expression occurs in proximal tubular epithelial cells, interstitial cells, glomerular parietal epithelial cells but can also be expressed by macrophages [55,56]. Upstream, iNOS expression and activation can be modulated by MAPK/NF-kB signalling pathways as well as JAK/STAT cascades stimulated by cytokines such as IFN-γ [30,57]. Furthermore, iNOS is implicated in cytoplasmic and macrophage ROS production [58], which can exacerbate SAA-induced inflammation.

Overall, iNOS^+^ DAB immunostaining was elevated in renal sections from the SAA group (Figure 8A) compared to the vehicle controls (Figure 8B). In general, iNOS^+^ DAB immunostaining was localised to the tubular epithelial cells as well as in the parietal epithelium lining the inner surface of Bowman’s capsule within glomeruli, particularly in the SAA group (Figure 8B; refer to black arrows). The extent of iNOS^+^ DAB immunostaining (Figure 8C) showed that iNOS expression in the parietal epithelium was ~1.9-fold greater in sections from the SAA group compared with controls (*p* < 0.001). Activated glomerular parietal epithelial cells in focal segmental glomerulosclerosis can cause the parietal epithelial cells to migrate to the glomerular tuft, producing matrix proteins, which induces scarring, fibrosis and, possibly, renal dysfunction in the long term. These results would suggest that SAA-induced renal inflammation may lead to the activation of the aforementioned cascades in upregulating iNOS expression, which can lead to fibrosis and renal dysfunction.

### 6.8. SAA Administration Leads to Fibrotic Changes in the Kidneys

Renal fibrosis is a progressive pathological result of extracellular matrix accumulation and the scarring of tissue in which connective tissue structures, such as collagen, take the place of normal parenchyma. Available evidence indicates that p38 MAPK activation and iNOS expression is correlated with renal fibrosis [59,60]. As both these markers were upregulated in this study, we next assessed renal fibrosis, which increased 16 weeks after cessation of SAA treatment as judged by the extent of PSR^+^-collagen staining localised around the glomeruli in the renal cortex and interstitial cells in the medulla (Figure 9A; refer to black arrows).

To assess group differences for PSR staining with greater detail, images were quantified using Fiji software (ImageJ, version 2.0). In the cortical region, PSR staining (Figure 9B) was ~2.2-fold greater in renal sections from the SAA group than with vehicle controls (*p* < 0.05). In the medulla, PSR staining (Figure 9C) was ~1.9-fold greater in the SAA group compared with controls (*p* < 0.05). Together, an acute SAA-induced inflammatory response could stimulate an oxidative stress pathway leading to macrophage activation, which in the long term, leads to the deposition of collage and fibrosis in the kidney in the chronic model. 

### 6.9. SAA Administration Causes Atherosclerotic Lesions Development in the Mouse Aortae

Renal dysfunction and renal disease are a major risk factor for atherosclerosis and cardiovascular disease [61]. Atherosclerotic lesions are common in the aortic valve in mice, which is a tricuspid semilunar valve consisting of three leaflets situated between the left ventricle and the aorta [62]. Aortic sections obtained 16 weeks after cessation of SAA treatment were stained with PSR and imaged, with lesions outlined for both control (Figure 10A; refer to black arrows) and SAA groups (Figure 10B; black arrows). When quantified, the percentage of lesion area was ~2.2-fold greater in aortic sections from the SAA group (Figure 10C) compared with sections from the control group (*p* < 0.05).

Further investigation revealed key differences in specific histological features of the atherosclerotic lesions between the treatment groups. Sections of lesions from the control group (refer to the green polygon were magnified; Figure 10D) indicated the presence of foam cells (Figure 10D; red arrow) and large cholesterol clefts (Figure 10D; green arrow) with a thick and distinctive fibrous cap (Figure 10D; blue arrow) consistent with ApoE^−/−^ mice being prone to atherosclerotic lesion development. In the corresponding SAA group (Figure 10E), a similar well-defined fibrous cap was observed. However, the periphery of the fibrous cap (Figure 10E; blue arrow) was not clearly demarcated, and the margin showed evidence of infiltration by foam cells, which were also present over the capsular structure (Figure 10E; red arrows). Plaque in mice from the SAA group showed foam cells within the lesion core and as well as several sharp cholesterol clefts (Figure 10E; green arrows). These combined histological features are characteristic of unstable plaque, which has more severe clinical implications. These results suggest that SAA-induced inflammation exacerbates atherosclerotic lesion pathogenesis in the long term, and the phenotype of the lesions may also be more prone to rupture, thus increasing the risk for long-term cardiovascular disease.

## 7. Discussion

While the inflammasome provides all necessary conditions to resolve tissue damage and elicit healing processes, unresolved or overactive chronic inflammation can cause host tissue damage [6]. This duality emphasises the “double-edged” nature of inflammatory activity. Although the immune and inflammatory cascades are considered “conserved” responses, the exact signalling pathways are heterogenous and can vary depending on the type of insult or injury and on the impacted tissue [63]. As a result, finding novel therapeutic targets within the immune and inflammatory cascades needs to be attuned for each response pathway rather than the targeting of the cascades as a collective [10]. The role SAA plays in regulating these inflammatory cascades remains unclear. In this study, acute SAA administration induced an increase in renal inflammatory markers, which led to long-term histological alterations in renal and aortic tissue, synonymous with renal dysfunction and atherosclerotic lesion formation. Upstream, upregulation of oxidative stress related transcription factors Nrf2, p65 NF-kB and p38 MAPK led to downstream activation of an immunoregulatory response with an increase in IFN-γ content and iNOS expression, a cascade which culminated in the endpoints of accelerated renal fibrosis and atherosclerotic lesion formation within the aortic sinus in the long term.

### 7.1. SAA Induced Expression of Nrf2, NF-kB and Activated p38 MAPK in the Renal Tissue

The Nrf2 protein is constitutively expressed, but under normal physiological conditions, is ubiquitinated by the Keap1/Cu3 ubiquitin ligase [64]. Turnover of Nrf2 enables responses to the dynamic oxidative environment. Overproduction of damaging oxidants, particularly hydrogen peroxide, modifies the cysteine thiol substituents on the Nrf2/Keap1/Cu3 complex, which inhibits the degradation of Nrf2 protein, enabling translocation to the nucleus and the induction of ARE genes [43]. This is consistent with the current study, which shows Nrf2-positive immunostaining largely localised to the nuclei of tubular epithelial cells. The lack of Nrf2-positive immunostaining within the glomeruli is consistent with other in vivo murine models [65]. However, the lack of glomerular Nrf2 staining compared to the tubular cells may be attributed to the relatively high rate of adenosine triphosphate (ATP) production required for the active transport and filtration of solutes and water across the tubular epithelium, a process that requires abundant oxygen consumption by mitochondria, which also produces reactive oxygen species [66]. Accordingly, the results from this current study suggest that SAA administration leads to an imbalance of oxidants, with a disproportionally greater oxidative stress induced within tubular cells than cells of the glomeruli, where mesangial cells might provide better protection against oxidative stress [67]. 

Data from this study show both positive p-p65 NF-kB and Nrf2 immunostaining largely localised to the nuclei of the tubular epithelia, indicating transcriptional activation. However, the molecular crosstalk between the redox sensitive Nrf2 and NF-kB transcription factors and how this interaction may regulate oxidative responses induced by SAA is unclear. For example, both Nrf2 and NF-kB activity require binding to the CREB-binding protein (CBP) transcriptional coactivator. A reciprocating competition for CBP complex formation means that overexpression of one transcription factor drives down CBP availability to form complexes with the other and, therefore, competition for CBP could limit downstream responses [68]. However, the modulatory activity between Nrf2 and NF-kB can vary depending on cell type [69]. Further complicating this level of competition, Keap1 (the regulator of Nrf2 in the cytoplasm) can degrade IKKs, which prevents NF-kB activation [70]. Furthermore, the production of certain mediators during inflammation, such as cyclooxygenase-2 (COX-2), can increase Nrf2 activity while inhibiting NF-kB activation. The results from the current study show that both Nrf2 and NF-kB transcription factors were both upregulated, which is suggestive of a cooperative relationship between Nrf2 and NF-kB signalling within the renal tissue of SAA-treated mice. Notably, the GTP-binding protein RAC1 induces the effects of Nrf2 through NF-kB activation in RAC1-mediated inflammation [71], suggesting the complexity of the functional crosstalk between Nrf2 and NF-kB, which further supports the notion that crosstalk occurs between these activated transcription factors.

Along with NF-kB, SAA administration has been linked to p38 MAPK activation through binding the cluster of differentiation 36 (CD36) receptor [72]. Activation of p38 MAPK, like NF-kB, results in the production of pro-inflammatory cytokines such as TNF and COX-2 [49]. In kidneys, p-p38 MAPK is largely localised to the tubular cells and the interstitial myofibroblasts [46]. Accordingly, in SAA-treated mice, p-p38 MAPK was detected in the nuclei of tubular epithelial cells with a similar distribution to that of Nrf2 and NF-kB. The members of the MAPK family are well-established cofactors of NF-kB activity; Sakai et al. showed that p38 MAPK phosphorylation and NF-kB activity is closely associated in the pathophysiology of human crescentic glomerulonephritis [73]. However, the functional crosstalk between p38 MAPK and Nrf2 is not well established, with some studies showing p38 MAPK activity resulting in the phosphorylation of Nrf2, which enhances the binding of Nrf2 and Keap1 and prevents translocation to the nucleus [64], while others suggest that this phosphorylation only has a limited effect on Nrf2 activity [74]. In this study, with SAA-induced inflammation was a concomitant upregulation in the activity of renal p38 MAPK, NF-kB and Nrf2 transcription factors, indicating SAA stimulates a raft of inflammatory and stress markers implicated in renal dysfunction.

### 7.2. SAA Induced Expression of IFN-γ in the Renal Tissue

The pro-inflammatory IFN-γ cytokine is predominantly synthesised by T cells and NK cells, inducing a plethora of immunomodulatory responses. IFN-γ signalling occurs through its binding with its cell-surface receptor (IFN-γR), resulting in its internalisation and nuclear accumulation [75]. It is also reported that the IFN-γ/IFN-γR interaction is responsible for the translocation of the STAT1 transcription factor to induce the transcription of IFN-γ genes [76]. In the present study, IFN-γ^+^ immunostaining was localised predominantly in the nuclei of renal tubular cells, with minor cytoplasmic and glomerular staining. NF-kB stimulation upon T helper type 1 cell activation has also been shown to upregulate IFN-γ production [77], which then amplifies NF-kB activity, forming a bidirectional positive feedback loop [78]. Furthermore, IFN-γ can exacerbate oxidative imbalances through ROS production and NADPH oxidase (NOX) upregulation [79], propagating oxidative stress responses. In that sense, SAA-induced oxidative stress may induce an immunostimulatory pathway in which oxidative imbalances are sustained. One of the major effects of IFN-γ is the priming of the immune system, such as augmenting TLR-mediated responses, which in turn enhance p38 MAPK/NF-kB signalling and macrophage activation [80]. Additionally, IFN-γ expression by dendritic cells and macrophages have revealed an autoactivation pathway for frontline acute responses, further locally enhancing SAA-induced oxidative stress [81]. While previous studies with SAA have demonstrated the ability of SAA to upregulate renal IFN-γ levels [19], the current study expands on this observation to indicate that upregulation of renal IFN-γ may result from SAA-induced oxidative stress pathways involving an immunostimulatory loop response through p38 MAPK/NF-kB signalling.

### 7.3. SAA Induced Fibrotic Changes in the Renal Tissue

In the present study, we have identified histopathological evidence for increased and more diffuse fibrotic and collagen depositions within the renal tissue of the chronic murine model. Overall, fibrosis was localised around the glomeruli and within interstitial spaces of the kidney. Fibrotic depositions around the glomeruli may be explained through the deposition of extracellular matrix by myofibroblasts derived from either glomerular epithelial or macrophage transition [58,82]. It has also been suggested that ROS-induced renal fibrosis upregulates renal NOX and superoxide radical anion formation, which induces fibroblast activation [83]. In lung tissue, upregulated p-p38 MAPK has been linked to fibrosis, with the implication that p-p38 MAPK activates fibrotic responses in fibroblasts [84,85]. Furthermore, p38 MAPK activation may be associated with renal damage through disrupting tight junctions between tubular epithelial cells [86]. Notably, the pattern of fibrosis in the current study showed colocalised p-p38-positive cells at the glomerular margins where fibrosis was evident, which provides evidence to link SAA-stimulated MAPK activation with fibrosis. 

In addition to the primary consequences of renal fibrosis affecting kidney function, renal fibrosis and renal dysfunction can lead to cardiovascular disease [87]. Renal dysfunctions may contribute to endothelial dysfunction and the development of atherosclerotic lesions [88]. Moreover, chronic kidney disease also induces systemic inflammation, potentially exacerbating the process of atherosclerosis [89]. In the chronic murine model in the current study, the atherosclerotic lesion area was significantly greater in the SAA group compared to controls. This result is consistent with a previous study demonstrating that SAA significantly increased aortic root lesion area in a similar model [19]. Lesion formation was also detected in the vehicle controls due to the utilisation of the ApoE^−/−^ model, which is prone to atherosclerotic lesion development even in the absence of a Western diet [90].

### 7.4. SAA Induced Atherosclerotic Changes in the Aorta Root of Mice

Atherosclerotic lesion advancement and complexity was shown to be a more important prognostic feature than lesion area [91]. For example, unstable fibrous plaques in diabetics yield poorer survival outcomes as there is a higher propensity for heart failure in these patients [92]. Histopathologic features of lesions from the SAA group presented with relatively higher cholesterol clefts, which damage and undermine fibrous cap integrity, eventually leading to acute cardiac events [93]. Fibrous caps in lesions from the SAA group lacked clear peripheral margins due to infiltrating foam cells and a lower density of cells forming the cap, signs of a clinically advanced lesion [94]. Therefore, SAA not only induces accelerated aortic lesion formation but, also, the lesion composition induced by SAA may represent a worse prognosis for acute events and thus highlights a need to prevent inappropriate chronic extension of SAA-induced oxidative stress.

## 8. Conclusions

Taken together, our results indicate that SAA induces inflammatory cascades potentially via ROS-mediated pathways that have significant impact not only on renal but vascular function, as summarised in Figure 11. Accordingly, the specific involved ROS and the oxidative damage that these ROS elicit need to be carefully identified and characterised for consideration as a therapeutic target and part of the signalling circuitry in SAA-induced inflammation, rather than a product of the pro-inflammatory cascade. This heterogeneity and complexity within ROS signalling, and its amplification of pathophysiological cascades when inappropriately extended, may explain the lack of clinical success in previous intervention studies with antioxidants. As such, even greater detail of SAA-induced ROS sources is required for their therapeutic targeting, with a requirement to identify and characterise the domains of ROS production for specific delivery and therapeutic targeting.

## Figures and Tables

**Figure 1 ijms-22-12582-f001:**
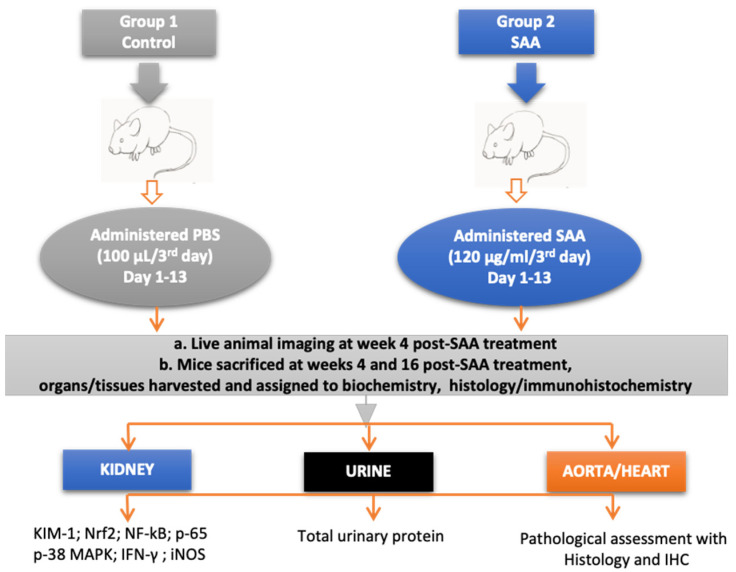
Schematic figure summarising the experimental design and analytical approach to assess tissue isolated from control and SAA-stimulated mice.

**Figure 2 ijms-22-12582-f002:**
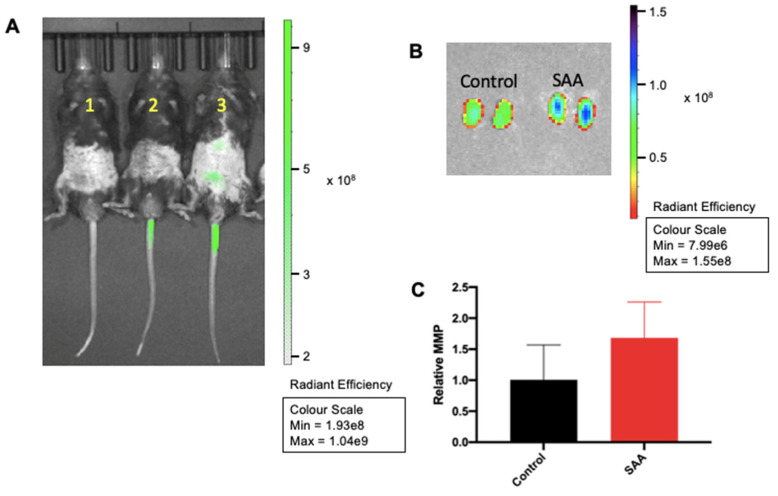
SAA administration activates MMPsense activity in vivo. Male ApoE^−/−^ mice were randomly allocated to vehicle control (administered 100 µL PBS every 3 days for 2 weeks) and the SAA group (administered 12 µg SAA protein every 3 days for 2 weeks). MMPsense (2 nmol–150 uL) was injected via tail vein 24 h prior and animals were imaged using IVIS^®^ SpectrumCT (PerkinElmer). (**A**) In vivo mouse images after MMPsense injection (n = 3/group); mouse (1) control without MMP injection; (2) control with MMPsense injection and (3) SAA-treated with MMPsense injection. (**B**) Representative images of isolated kidneys from control and SAA group. After imaging, animals were sacrificed and organs harvested. Kidneys were isolated and imaged using IVIS^®^ SpectrumCT (PerkinElmer) for MMS sense activity. (**C**) MMPsense signal intensity in mice images was quantified using Living Image^®^ (PerkinElmer) data analysis software.

**Figure 3 ijms-22-12582-f003:**
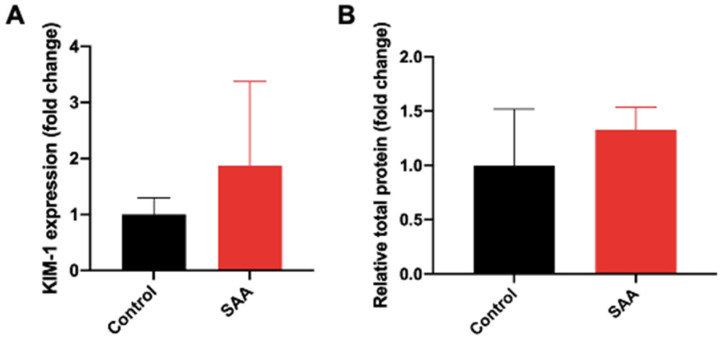
Assessment of renal function following SAA administration. (**A**) Male ApoE^−/−^ mice were randomly allocated to vehicle control (administered 100 µL PBS every 3 days for 2 weeks) and the SAA group (administered 12 µg SAA protein every 3 days for 2 weeks). Kidney tissue was harvested 4 weeks after cessation of treatment, and a portion of renal tissue was homogenised for biochemical analyses. An immunoassay ELISA kit to quantify renal KIM-1 expression was utilised to assess renal dysfunction. Reagents, standards and samples were prepared and assayed by ELISA as per the manufacturer’s instructions (Abcam). Absorbance values were measured at 450 nm with values for concentration calculated from the standard curve generated. (**B**) Total urinary protein concentration; all data shown as relative mean ± SD.

**Figure 4 ijms-22-12582-f004:**
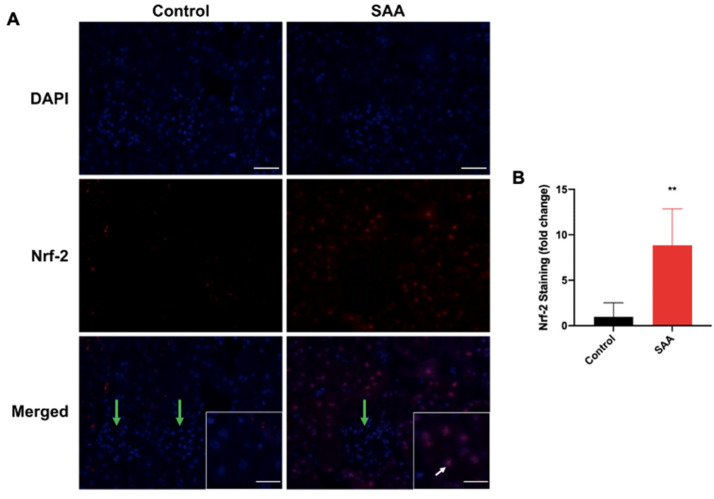
SAA administration stimulates Nrf2 expression in the renal cortical tissue. (**A**) Male ApoE^−/−^ mice were randomly allocated to vehicle control (administered 100 µL PBS every 3 days for 2 weeks) and the SAA group (administered 12 µg SAA protein every 3 days for 2 weeks). Kidney tissue was harvested 4 weeks after cessation of treatment and fixed in situ before embedding and sectioning (5 µm). Renal sections were dewaxed and rehydrated before undergoing heat-induced antigen retrieval. Cortical Nrf2 expression was assessed using immunofluorescence microscopy. Slides were visualised at 40× magnification (scale bar = 20 μm); images are representative of at least 4 fields of view for each sample. Nuclei were stained with DAPI (blue) and Nrf2 with an appropriate Opal fluorophore (red). White arrows show regions of relatively high tubular Nrf2+ immunostaining. Green arrows show regions of relatively low glomerular Nrf2+ immunostaining. Insets show higher magnification images (scale bar = 10 μm), Nrf2+ staining was mixed with Nrf2 colocalised to nuclei with residual cytoplasmic staining. Representative images show cortical fields from n = 5 (Control), n = 6 (SAA). (**B**) Immunostaining was quantified using a mean staining intensity for each field of view and averaged for each sample. Data shown as relative mean ± SD. ** Relative to control group; *p* < 0.05.

**Figure 5 ijms-22-12582-f005:**
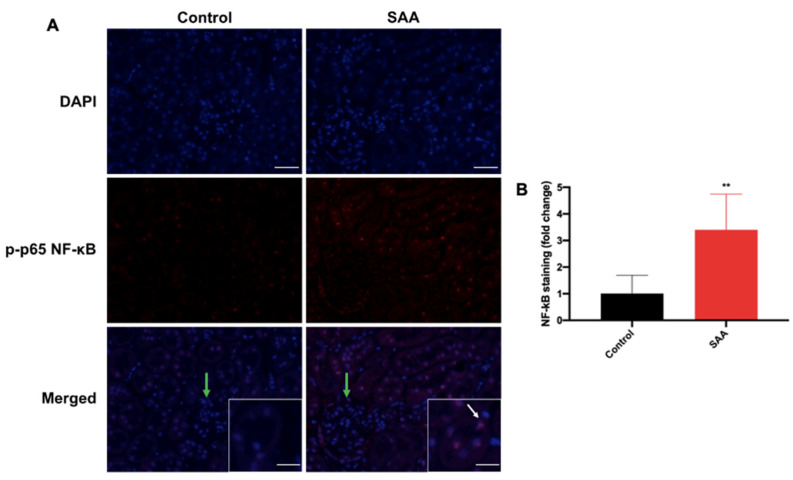
SAA administration stimulates NF-kB p-p65 in the renal cortical tissue. Male ApoE^−/−^ mice were randomly allocated to vehicle control (administered 100 µL PBS every 3 days for 2 weeks) and the SAA group (administered 12 µg SAA protein every 3 days for 2 weeks). Kidney tissue was harvested 4 weeks after cessation of treatment and fixed in situ before embedding and sectioning (5 µm). Renal sections were dewaxed then rehydrated before undergoing heat-induced antigen retrieval. (**A**) NF-kB p-p65 expression was assessed using immunofluorescence microscopy. Slides were visualised at 40× magnification (scale bar = 20 µm); images are representative of at least 4 fields of view for each sample. Nuclei were stained with DAPI (blue) and NF-kB p-p65 with an appropriate Opal fluorophore (red). White arrows show NF-kB+ staining localised to renal epithelial cells and not in the glomerular endothelium (glomeruli indicated by green arrow). Insets show higher magnification (scale bar = 10 µm) images of renal tubular epithelial cells with NF-kB p-p65+ staining largely colocalised to nuclei with some residual cytoplasmic staining. Representative images show cortical fields from n = 5 (control), n = 6 (SAA). (**B**) Immunostaining was quantified using a mean staining intensity for each field of view and averaged for each sample. Data shown as relative mean ± SD. ** Different to the control group; *p* < 0.001.

**Figure 6 ijms-22-12582-f006:**
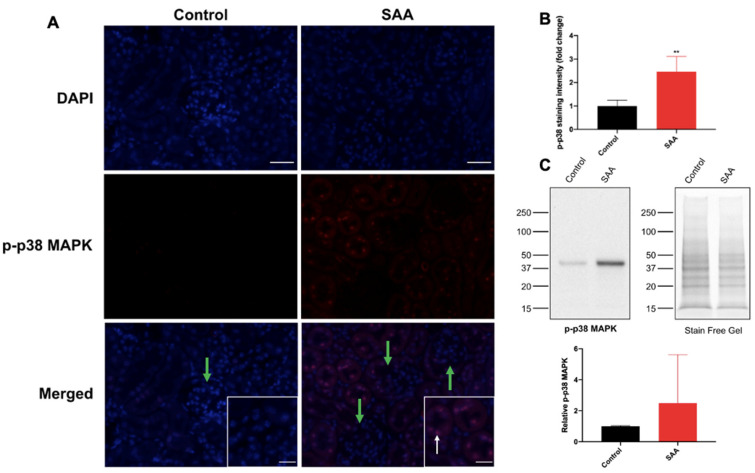
SAA administration stimulates p-p38 MAPK expression in the renal cortical tissue. (**A**) Male ApoE^−/−^ mice were randomly allocated to vehicle control (administered 100 µL PBS every 3 days for 2 weeks) and the SAA group (administered 12 µg SAA protein every 3 days for 2 weeks). Kidneys were harvested 4 weeks after treatment cessation and fixed in situ before embedding and sectioning (5 µm). Renal sections were dewaxed then rehydrated before undergoing heat-induced antigen retrieval. NF-kB p-p65 expression was assessed using immunofluorescence microscopy. Slides were visualised at 40× magnification (scale bar = 20 µm); images are representative of at least 4 fields of view for each sample. Nuclei were stained with DAPI (blue) and p-p38 MAPK with an appropriate Opal fluorophore (red). White arrow indicates p-p38 MAPK+ staining localised to renal epithelial cells, which was absent in glomerular endothelium (glomeruli indicated by green arrow). Insets show higher magnification (scale bar = 10 µm) images of renal tubular epithelial cells with p-p38 MAPK+ staining colocalised to nuclei with some cytoplasmic staining. Representative images show cortical fields from n = 5 (control), n = 6 (SAA). (**B**) Immunostaining was quantified using a mean staining intensity for each field of view and averaged for each sample; data shown as relative mean ± SD. ** Different to control group; *p* < 0.001. (**C**) Western blot analyses of p-p38 MAPK were performed as described in the methods section. Homogenised renal tissue (20 µg protein) from the control and SAA groups were separated by SDS-PAGE. Proteins were transferred onto a membrane, blocked, then incubated with the appropriate antibodies. Membranes were imaged, and bands at 38 kD corresponding to p-p38 MAPK were identified and quantified using densitometry (ImageLab, version 6.0.1). All density data were normalised with total protein loading determined from corresponding stain free gel images. Data shown as relative mean ± SD.

**Figure 7 ijms-22-12582-f007:**
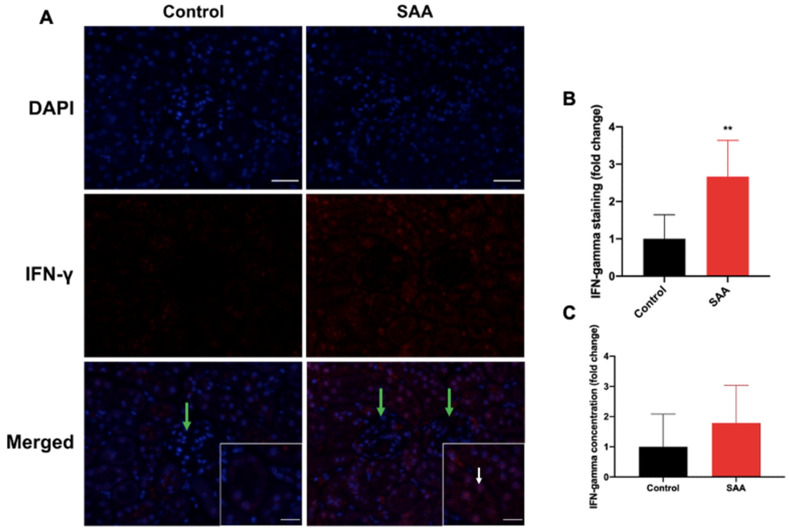
SAA administration stimulates IFN-γ expression in the renal cortical tissue. (**A**) Male ApoE^−/−^ mice were randomly allocated to vehicle control (administered 100 µL PBS every 3 days for 2 weeks) and the SAA group (administered 12 µg SAA protein every 3 days for 2 weeks). Kidney tissue was harvested 4 weeks after cessation of treatment and fixed in situ before embedding and sectioning (5 µm). Renal sections were dewaxed then rehydrated before undergoing heat-induced antigen retrieval. IFN-γ localisation was assessed using immunofluorescence microscopy. Slides were visualised at 40× magnification (scale bar = 20 µm); images are representative of at least 4 fields of view for each sample. Nuclei were stained with DAPI (blue) and IFN-γ with an appropriate Opal fluorophore (red). White arrow indicates IFN-γ staining primarily localised to renal epithelial cells and minimally in the glomerular endothelium (glomeruli indicated by green arrow). Insets show higher magnification (scale bar = 10 µm) images of renal tubular epithelial cells with IFN-γ staining largely colocalised to nuclei with some residual cytoplasmic staining. Representative images show cortical fields from n = 5 (control), n = 6 (SAA). (**B**) Immunostaining was quantified using a mean staining intensity for each field of view and averaged for each sample. Data shown as relative mean ± SD. ** Different to the control group; *p* < 0.001. (**C**) An ELISA kit was utilised to quantify renal IFN-γ as described in the methods section. Results shown as relative mean ± SD.

**Figure 8 ijms-22-12582-f008:**
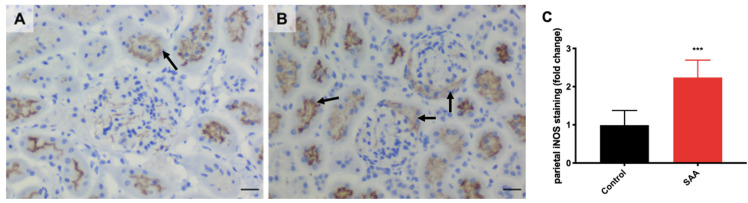
SAA administration stimulates iNOS expression in the renal cortical tissue. Male ApoE^−/−^ mice were randomly allocated to (**A**) the vehicle control group (administered 100 µL PBS every 3 days for 2 weeks) and (**B**) the SAA group (administered 12 µg SAA protein every 3 days for 2 weeks). Kidney tissue was harvested 4 weeks after cessation of treatment and fixed in situ before embedding and sectioning (5 µm). Renal sections were dewaxed then rehydrated before undergoing heat-induced antigen retrieval. iNOS expression was assessed using immunohistochemistry and light microscopy. Slides were visualised using an Axio Scope.A1 light microscope at 40× magnification (scale bar = 20 µm); images are representative of at least 4 fields of view for each sample. For all images shown, nuclei are stained with haematoxylin (appearing as blue), and iNOS with DAB (appearing as brown). Black arrows indicate iNOS+ immunostaining (DAB positive) localised to epithelial cell brush borders and to the parietal epithelial cells lining the inner surface of Bowman’s space in the SAA group. Representative images show iNOS+ immunostaining in renal cortical fields from n = 5 (control), n = 6 (SAA). (**C**) Parietal iNOS+ staining within the glomeruli was quantified using mean grey value and an optical density calculation for each field of view and averaged for each sample. Data shown as relative mean ± SD. *** Different to the control group; *p* < 0.0001.

**Figure 9 ijms-22-12582-f009:**
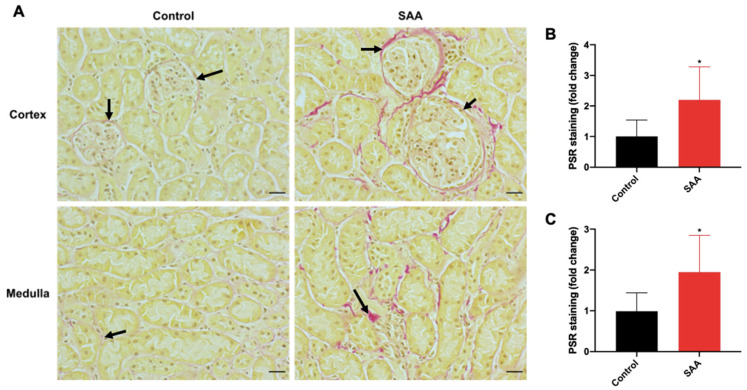
SAA administration causes long-term fibrotic changes in kidneys. (**A**) Male ApoE^−/−^ mice were randomly allocated to vehicle control (administered 100 µL PBS every 3 days for 2 weeks) and the SAA group (administered 12 µg SAA protein every 3 days for 2 weeks). Kidney tissue was harvested 14 weeks after cessation of treatment and fixed in situ before embedding and sectioning (5 µm). Renal sections were dewaxed then rehydrated before staining with haematoxylin and picrosirius red (PSR) solution. Renal fibrosis was assessed using the PSR stain for collagen. Slides were visualised using an Axio Scope.A1 light microscope at 40× magnification (scale bar = 20 µm); images are representative of at least 4 fields of view for each sample. For all images shown, nuclei are stained with haematoxylin (brown), and collagen with PSR (red). Black arrows indicate PSR staining localised to the parietal epithelial cells lining the outer surface of Bowman’s space and the interstitial spaces in the SAA group. Representative images show PSR+ staining in renal cortical and medullary fields from n = 8 (control), n = 6 (SAA). PSR staining in the (**B**) cortical and (**C**) medullary regions were quantified using thresholding tools for each field of view and averaged for each sample. Data shown as relative mean ± SD. * Different to the control group; *p* < 0.05.

**Figure 10 ijms-22-12582-f010:**
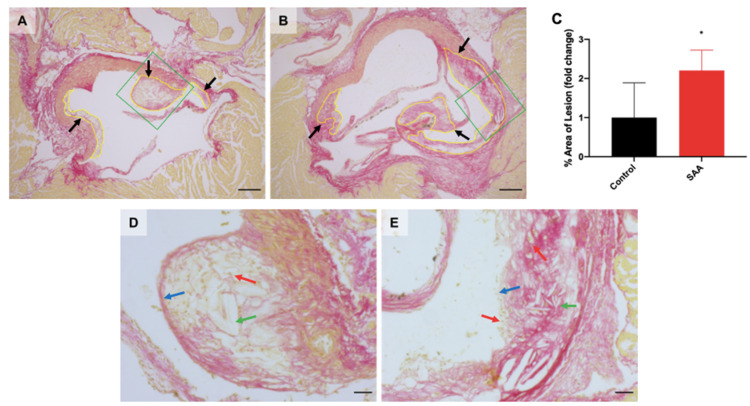
SAA administration causes atherosclerotic lesion development at aortic valve leaflets. Male ApoE^−/−^ mice were randomly allocated to (**A**) the vehicle control (administered 100 µL PBS every 3 days for 2 weeks) and (**B**) the SAA group (administered 12 µg SAA protein every 3 days for 2 weeks). The aortic sinus from each mouse was harvested 16 weeks after cessation of SAA treatment and fixed in situ before embedding and sectioning (5 µm). Aortic sections were dewaxed then rehydrated before staining with haematoxylin and picrosirius red (PSR) solution. Slides were visualised using an Axio Scope.A1 light microscope at 5× magnification (scale bar = 200 µm). For all images shown, cardiac muscle tissue is stained with haematoxylin (brown) and collagen with PSR (red). The black arrows and yellow outlines indicate lesion formation localised at the root of the valve leaflets. The green polygons indicate the field of view magnified at 20×. Representative images show atherosclerotic lesion formation from n = 5 (control), n = 7 (SAA). (**C**) The total lesion size for each sample was quantified using a freehand drawing tool (ImageJ, version 2.0) and calculated as percentage of total area. Data shown as relative mean ± SD. * Different to the control group; *p* < 0.05. High magnification (20×, scale bar = 40 µm) images of atherosclerotic lesions at the aortic roots in the vehicle control (**D**) and SAA groups (**E**) were obtained at the fibrous caps (blue arrows) where foam cells (red arrows) and cholesterol clefts (green arrows) can be visualised.

**Figure 11 ijms-22-12582-f011:**
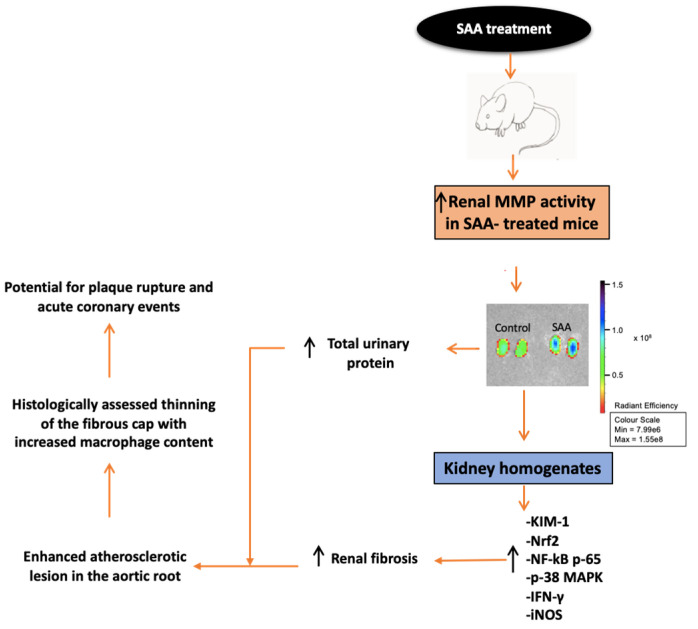
Schematic flow chart summarising the biological implications to kidney and vasculature in mice stimulated with pathological amounts of SAA that simulated an acute phase response over 2 weeks, with follow up at 4 and 16 weeks after cessation of the 2-week period of SAA administration in mice.

## Data Availability

Not applicable.

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
