# Peer review of "Pro-Inflammatory Serum Amyloid a Stimulates Renal Dysfunction and Enhances Atherosclerosis in Apo E-Deficient Mice"

_ijms, 2021, doi:10.3390/ijms222212582_

Round 1

Reviewer 1 Report

Dear Editor,

I evaluated the paper from Gao et al. and I really appreciated their work. The exact identification of the molecular mechanisms by which amiloid can exert its negative role in human tissues is fundamental of the possible finding of new treatments to counteract the disease. The paper is well written. The English is fluent. The methods are good.

Author Response

We thank the reviewer’s time in reviewing the manuscript and the positive feedback on our study.

Reviewer 2 Report

Dear Authors

The article sent for my review is very interesting and will definitely interest the readers of IJMS. It concerns research conducted to understand the effects of acute serum amyloid A  on renal dysfunction and the severity of atherosclerosis in apo E deficient mice. The article is fair, and you can see an excellent research technique of the authors.

The title and abstract cover the main aspect of the article and are appropriate.

The introduction provides background and information relevant to the study.

However, the purpose is somewhat hidden. In the last sentence of the introduction, the authors wrote only: "In the present study, inflammation was induced by SAA administration to assess the key players involved in renal inflammation as well as their long-term endpoints." I believe that it should be more extensive.

The methods are precise and reproducible and all results presented are consistent with the methods described in the manuscript.
This is where a flow diagram would come in handy, making it easier for a potential reader to follow the researchers.

The results are innovative and interestingly presented, and the study is a progress in the field of understanding the pathomechanisms of renal Dysfunction and Enhancement of Atherosclerosis. 

In the discussion, the commentary to the results described by the authors correlates with their obtained results. The discussion is extensive, and this is, on the one hand, its plus, because the authors try to explain everything thoroughly and also present their doubts; on the other hand, it is pretty challenging to read. In my opinion, it would be helpful to break the discussion down into subsections.

Moreover, there are no conclusions or summaries in the manuscript, which is the most significant weakness of this manuscript. I know that the IJMS does not require it, but with such extensive work, a summary of the obtained results would be helpful. The best will be a summary along with the figure of a proposal on how a studied process can proceed.

Tables and figures are clear and legible. The drawings seem to be free from unnecessary modifications.
The manuscript does not raise any ethical concerns. The statistics are appropriate.

To sum up, the article is very good but quite complicated to get through, so my suggestions for modifications may make it easier to understand the presented content.

Best regards

Author Response

Dear Authors

The article sent for my review is very interesting and will definitely interest the readers of IJMS. It concerns research conducted to understand the effects of acute serum amyloid A  on renal dysfunction and the severity of atherosclerosis in apo E deficient mice. The article is fair, and you can see an excellent research technique of the authors.

The title and abstract cover the main aspect of the article and are appropriate.

The introduction provides background and information relevant to the study.

  1. However, the purpose is somewhat hidden. In the last sentence of the introduction, the authors wrote only: "In the present study, inflammation was induced by SAA administration to assess the key players involved in renal inflammation as well as their long-term endpoints." I believe that it should be more extensive.

As requested, we have added the additional text to the revised manuscript (refer to page 3, paragraph 1 lines 15-18 of the revised manuscript). The new text reads as follows:

“In the present study, inflammation was induced by SAA administration to investigate the role for SAA in promoting renal and vascular dysfunction in a murine model. Herein, we further elucidate the key players involved in renal inflammation as well as their long-term endpoints”.

The methods are precise and reproducible and all results presented are consistent with the methods described in the manuscript. 2. This is where a flow diagram would come in handy, making it easier for a potential reader to follow the researchers.

As suggested by the reviewer we have added a flow diagram as new figure 1 in the revised version of the manuscript. All other figures citations in the text have been adjusted to accommodate this additional schematic figure.  We also added an explanation for the use of two groups of mice at different time points as outlined on page 3, Paragraph 2 and lines 10-16 of paragraph 2. The new figure 1 is on page 4 in the revised manuscript. The new text reads as follows”

“The experimental design together with the planned analyses is summarised schematically in Figure 1. Separate groups of control and SAA treated mice were monitored for 4 or 16 weeks after cessation of SAA stimulation and organs were harvested for biochemical and histological/immunohistochemical analyses at these times.  Notably, these time points were selected to mimic the earliest stages prior to marked development of atherosclerotic lesion and then at the end point were more severe atherosclerosis is noted in ApoE-deficient mice”.

The results are innovative and interestingly presented, and the study is a progress in the field of understanding the pathomechanisms of renal Dysfunction and Enhancement of Atherosclerosis. 

In the discussion, the commentary to the results described by the authors correlates with their obtained results. The discussion is extensive, and this is, on the one hand, its plus, because the authors try to explain everything thoroughly and also present their doubts; on the other hand, it is pretty challenging to read.

  1. In my opinion, it would be helpful to break the discussion down into subsections.Moreover, there are no conclusions or summaries in the manuscript, which is the most significant weakness of this manuscript. I know that the IJMS does not require it, but with such extensive work, a summary of the obtained results would be helpful. The best will be a summary along with the figure of a proposal on how a studied process can proceed.

As suggested by the reviewer we have divided the discussion into sub sections and also added a summary flow chart (refer to new figure 11 on page 19 in the revised manuscript) that defines and highlights all the key experimental outcomes of the study.

Tables and figures are clear and legible. The drawings seem to be free from unnecessary modifications.  The manuscript does not raise any ethical concerns. The statistics are appropriate.  To sum up, the article is very good but quite complicated to get through, so my suggestions for modifications may make it easier to understand the presented content.

We thank the reviewer for these general positive comments.

Reviewer 3 Report

This study was well-prepared. The paper can be reconsidered as following points:

  1. The amount or dose of SAA admitted can be changed (i.e., 0.5x, 1.5x) in the experiments. This reviewer would like to know the amount/dose-response to the outcomes.
  2. In Abstract, the expression “SAA-induced in” (in line 4) might be in a grammatical error.
  3. In Abstract, the expression “SAA-induced an” (in line 10) might be in a grammatical error.
  4. In Abstract, the expression “IN” (in line 12) might be mistyped.
  5. In Abstract, the expression “;” (in line 15) might be difficult to understand the meaning of authors.
  6. In Abstract, the expression “INOS” (in line 17) might be mistyped.
  7. Did the authors use properly the expressions “SAA-induced ROS”, “SAA-induced ROSs”, or “SAA-induced oxidative stress”?
  8. Can the responses to SAA treatment be suppressed? How is the idea of authors?
  9. The English check would be required academically.

Author Response

This study was well-prepared. The paper can be reconsidered as following points:

  1. The amount or dose of SAA admitted can be changed (i.e., 0.5x, 1.5x) in the experiments. This reviewer would like to know the amount/dose-response to the outcomes.

Thank you for the suggestions and we agree that it would be interesting to test a dose-response for the pro-inflammatory SAA in this animal model of atherosclerosis/renal damage; this would be a significant novel study however this was not the aim of the present study. We would now clarify that only one stock solution of SAA (120 µg/ml) was used here and the final dose administered to each mouse equates to 12 microg/SAA protein per injection per three-day interval to stimulate an acute response that lasts for 2 weeks in total.  This amount is considered to be a moderate level of the acute phase protein based on the literature and measured physio-pathological threshold in the blood plasma, which can increase in circulation up to 1000 microg/mL under significant inflammatory conditions such as diabetes, as indicated in the original version of the text.

2. In Abstract, the expression “SAA-induced in” (in line 4) might be in a grammatical error.

Thank you we have addressed this error.

3. In Abstract, the expression “SAA-induced an” (in line 10) might be in a grammatical error.

We have revised the sentence to address this error.

4. In Abstract, the expression “IN” (in line 12) might be mistyped.

We have addressed this typo.

5. In Abstract, the expression “;” (in line 15) might be difficult to understand the meaning of authors.

We have replaced “expression” with “immunolocalization” to be more instructive

6. In Abstract, the expression “INOS” (in line 17) might be mistyped.

Thank you we have corrected this typo

7. Did the authors use properly the expressions “SAA-induced ROS”, “SAA-induced ROSs”, or “SAA-induced oxidative stress”?

We have used ROS where it refers to the production of reactive oxygen species and oxidative stress which refers global increases in the cellular/tissue levels of oxidants that can cause oxidative damage to the host tissue.

8. Can the responses to SAA treatment be suppressed? How is the idea of authors?

We have previously studied the inhibition of SAA induced pro-inflammatory makers in ex vivo models using cyclic nitroxides a treatment agent (Martin et al; Int. J. Mol. Sci. 2021, 22(9), 4549). However, the in vitro cellular response to a treatment does not always yield similar outcomes in vivo, thus we are interested to see how different treatment options respond in mouse models that could be more relevant to human model.  This has now been added to the conclusion (refer to Section 8 of the revised discussion on page 19) to emphasise the importance of understanding the molecular pathways for SAA stimulated inflammation in order to design optimal therapeutic drugs to target these pathways.

Round 2

Reviewer 3 Report

The paper was much improved, even if the native check may be still needed.